# Phenotypic Characteristics and Occurrence Basis of Leaf Necrotic Spots in Response of Weedy Rice to Imazethapyr

**DOI:** 10.3390/plants13091218

**Published:** 2024-04-28

**Authors:** Zeyu Zhang, Xianyu Wang, Jianing Zang, Dongsun Lee, Qian Zhu, Lijuan Chen

**Affiliations:** 1Rice Research Institute, Yunnan Agricultural University, Kunming 650201, China; zhangzeyuzzy99218@163.com (Z.Z.); dong_east@naver.com (D.L.); gabriel731@ynau.edu.cn (Q.Z.); 2The Key Laboratory for Crop Production and Smart Agriculture of Yunnan Province, Yunnan Agricultural University, Kunming 650201, China; 3State Key Laboratory for Conservation and Utilization of Bio-Resources in Yunnan, Yunnan Agricultural University, Kunming 650201, China

**Keywords:** weedy rice, imidazolinone herbicide, phenotype, leaf necrotic spots, abiotic stress

## Abstract

Weedy rice is the most challenging weed species to remove in rice production. We found a novel phenotype of seedling leaves which rapidly generates necrotic spots in response to imidazolinone herbicides in weedy rice, but its influencing factors and formation basis are still unknown. In this study, we used the leaf necrotic spot-producing type of weedy rice as the material. First, leaf necrotic spots were defined as physiological and vacuole-mediated cell necrosis by microscopic examination. The imazethapyr concentration was positively correlated with the degree of necrotic spots occurring, while the action site was in accordance with necrosis using herbicide stability tests combined with fluorescence parameters. Furthermore, transcriptome analysis revealed significant differences in the gene expression of endoplasmic reticulum stress and the lipid metabolism membrane structure damage pathway during necrosis, as confirmed by transmission electron microscopy. The light–temperature test also showed that high temperature and intense light could promote the appearance of necrotic spots. These experimental results are helpful in clarifying the process and basis of imazethapyr in inducing the rapid generation of necrotic spots in rice leaves and providing new insight into understanding the mechanism of response to imidazolinone herbicides and the control of weedy rice.

## 1. Introduction

Weedy rice (*Oryza sativa* f. *spontanea*) is a term used to describe rice with weedy characteristics. It is typically found in rice fields and their surrounding areas, serving as an intermediate species between wild and cultivated rice [1]. Most weedy rice produces mimicry, which closely resembles the morphology of cultivated rice. As a wild relative, it possesses stronger competitive advantages, disease resistance, and reproductive characteristics, such as early maturity, strong grain-shattering, and long dormancy [2,3]. The combination of these traits makes weedy rice the most challenging weed to manage in paddy fields [4]. The expansion of direct rice seeding areas has increased the incidence of weedy rice, causing the damage to expand annually [5,6]. To tackle this issue, breeders have developed several mutagenic or transgenic herbicide-resistant rice varieties. Among these, imidazolinone (IMI)-resistant varieties are the most mature and widely used due to their significant effect and easy accessibility through mutagenesis [7].

IMI herbicides cause a deficiency in the synthesis of branched-chain amino acids (BCAAs) by binding to the substrate entry and exit channels of the target protein, acetolactate synthase (ALS) [8]. In contrast, resistant rice varieties have undergone single nucleotide polymorphism (SNP) mutations that reduce the sensitivity of ALS [9]. This makes them ideal for use in breeding efforts for herbicide-resistant rice varieties, as they do not involve an active centre and impose little change to fitness [10]. However, due to their single-copy nature and low reproductive isolation, they are highly susceptible to resistant SNP drift [11]. Singh et al. studied 89 plots and 695 copies of weedy rice material in Arkansas, USA, of which, only 20% were sensitive, and the rest were resistant types and predominantly drifted, with a small percentage of mutant types [12]. The increasing problem of weedy rice escapes was almost ubiquitous in areas where Clearfield^®^ was used [13]. The most effective approach to reducing the frequency of resistant weedy rice is through integrated control using multiple methods to improve IMI herbicide efficacy and reduce selection pressure and weedy rice residues [14].

IMI herbicides, such as imazamox and imazethapyr, are commonly used in rice fields, and most weedy rice shows greening and yellowing and a slow damage process after application. Following herbicide application, weedy rice growth ceases within a few hours. The damage phenotype typically appears after 1–2 weeks, with meristematic tissues fading from green to yellow, followed by leaf wilting or necrosis from the tips. Visible damage is often evident after 3–4 weeks [15]. In our laboratory, we identified and evaluated imazethapyr resistance using a worldwide weedy rice group. We observed that some weedy rice showed necrotic spots on the leaves within a few days after herbicide application. This phenotype progressed rapidly and was affected by the weather. This observation has not been reported before. Although IMI herbicides have the advantages of a long persistence period and broad-spectrum activity, their drawbacks are also apparent. These include their mild temperament, slow onset of action, and high residual amounts of weedy rice. This study focused on leaf necrotic spot occurrence and mechanisms in weedy rice after imazethapyr application. We initially identified the spots based on appearance, cytology, and photosynthetic fluorescence parameters and investigated their population distribution. Then, the technical product and preparation stability tests were combined with fluorescence parameters to determine the correlation between necrotic spot occurrence and imazethapyr. An exploration of the basis of necrotic spots was carried out using electron microscopy and transcriptome analysis. Finally, the influence of environmental factors was determined by temperature and light treatment, and the cause of occurrence was verified. This study aimed to elucidate the relationship between imazethapyr and the underlying occurrence basis of necrotic spots, which will facilitate the development of a novel strategy for enhancing the efficacy and prolonging the service life of IMI herbicides in rice varieties.

## 2. Results

### 2.1. Phenotypic Identification of Leaf Necrotic Spots in Weedy Rice

Following the application of IMI herbicides to weedy rice with the code name HRT1, necrotic spots were observed primarily at the bends of the leaf blades of new leaves in the young seedling stage. These spots appeared within 3–5 days on sunny days after imazethapyr application. Initially, the spots appeared reddish-purple in colour and subsequently developed necrosis at the centre or directly produced necrosis without any colouration. In severe cases, the necrosis spread up and down, resulting in a large area of necrosis (Figure 1A,D). The phenotype of HRT1 differed significantly from the traditional phenotype of HRT2 (Figure 1B). HRT1 and HRT2 exhibited similar levels of tolerance and were grouped in the middle and upper parts of the population. HRT1 displayed the necrotic spotting phenotype, with the first spotting appearing after one week of herbicide treatment and no significant chlorosis (necrotic spot phenotype). In contrast, HRT2 began to fade green and yellow from the tip of the leaves after two weeks of herbicide treatment and gradually wilted (conventional phenotype). After 7 days of treatment, population identification (Figure 1C and Appendix A) showed that among the 113 weedy rice materials, the traditional phenotype accounted for 5% and the necrotic spotting phenotype accounted for 22%, with the necrotic spotting phenotype outnumbering the conventional phenotype, while the proportion of the conventional phenotype (43%) to the mixed phenotype (46%) increased sharply after 14 days, with the necrotic spotting phenotype mostly converting to the mixed phenotype, with a decreasing proportion (4%).

This type of necrotic spot had some similarities to the spots caused by the brown spot pathogen. However, by observing the leaf surfaces and microscopic examination (Figure 1D,E), the necrotic spots did not show obvious bacterial production of pus and no phenomenon of bacterial proliferation, which excluded the possibility of bacteria being the main cause of occurrence. Transmission electron microscopy results (Figure 1F) showed that the cells within the spots had obvious features of vacuole-mediated necrosis, such as cytoplasmic swelling, rupture of the vacuole membrane, and mixing of the cytoplasmic matrix.

### 2.2. Correlation Analysis of Leaf Necrotic Spot Formation and Imazethapyr in Weedy Rice

#### 2.2.1. Correlation between the Preparation and Formation of Necrotic Spots

The necrotic phenotype was mostly generated from the bends of the leaves. The smear experiment conducted on seedlings during the three-leaf stage showed (Figure 2A) that necrotic spots first appeared in the middle of the third leaf after application of the second leaf of HRT1 at the recommended dose of the formulation, excluding the effect of additives on the disruption of the waxy layer, leading to light scorching. To clarify the relationship between necrotic spots and formulation, we treated the resistant variety (Jinjing 818) with multi-level gradient spraying of the formulation. We found that the necrotic spot phenotype was also observed in the resistant variety when the concentration was up to 300 g a.i. ha^−1^ (Figure 2B). This suggests that necrotic spots can occur in resistant varieties when the herbicide concentration is sufficiently high. Combined with the characteristics of imazethapyr, including easy gathering in the growing area and still resistant varieties, the non-smear area and inhibitory resistance could trigger these necrotic spots, which initially showed a certain correlation between the occurrence of necrotic spots and imazethapyr.

#### 2.2.2. Necrotic Spots Can Be Triggered by Imazethapyr and Promoted by Auxiliary Agents

To determine the correlation between imazethapyr and necrotic phenotypes, we conducted comparative experiments on the efficacy of the technical product and the formulation. Photosynthetic fluorescence parameters are effective tools for evaluating the mode of action and effects of herbicides. Therefore, the simultaneous observation of these parameters was carried out to identify them accurately. The results showed that (Figure 3) the maximum photosynthetic efficiency (Fv/Fm), the actual photosynthetic efficiency (Fv’/Fm’), and the chlorophyll index (Chldx) of HRT1 decreased with an increase in the concentration of the technical product. Additionally, the quantum yield of the non-regulated energy dissipation (Y_NO_) increased, while the photo-quantum yield of the regulated energy dissipation (Y_NPQ_) did not change significantly. The RGB plots (Figure 3 and Figure 4A) showed that plant growth was not completely inhibited, except for the 1x formulation recommended dose (1x AS) and the 8x technical product recommended dose group (8x TC). The photosynthetic fluorescence and growth were hindered but showed an upward trend. The effects of the 1x AS group were the same as those of the 8x TC group. The correlation between each index and herbicide concentration and time was calculated. The best correlation was found with Fv/Fm (Appendix A).

Observation of the phenotype map and corresponding changes in the Fv/Fm fluorescence map of the 1x AS group (Figure 4A) showed that the necrotic spots were consistent with the area of decrease in Fv/Fm and stabilised at the bends of the leaves. However, the effect of 1x TC was not obvious; thus, the Fv/Fm of the original group on the third day was used for comparison. The decrease in Fv/Fm at the leaf bend increased with the increase in technical product concentration, and the Fv/Fm of 4x TC and 8x TC at the leaf bend was similar to that of 1x AS, indicating that Fv/Fm was negatively correlated with necrotic spots (Figure 4B,C). The occurrence of necrotic spots was positively correlated with the concentration of the technical product (Figure 4C (2), R = 0.949), and the occurrence of necrotic spots on the seventh day of 1x AS was in the middle of and significantly different from that of 2x TC and 4x TC. In conclusion, necrotic spotting was independently induced by imazethapyr and promoted by the formulation. The efficacy of 1x, the recommended dose formulation, was equivalent to 4x, the technical product dose of the treatment, and the technical product treatments at 2x and below were not sufficient to kill weedy rice.

In addition, according to the fluorescence results (Figure 3), Fv/Fm was less than 0.83. Although the control group rose steadily, it did not reach more than 0.8. Y_NPQ_ and was stable and less than 0.1, and Y_NO_ decreased but the overall decrease was about 0.4. This indicates that HRT 1 is in a state of photoinhibition or photodestruction in the seedling stage and that HRT 1 material cannot adapt to the light intensity of Kunming in the seedling stage. Combined with the high altitude of Kunming, the light intensity was twice as high as that of plain areas; thus, it is speculated that intense light may be one of the reasons for the occurrence of necrotic spots.

### 2.3. Transcriptome Analysis Showed Differential Expression of Endoplasmic Reticulum Injury Stress

To investigate the mechanism by which imazethapyr triggers necrosis, we examined the transcriptional status of HRT1 following treatment with the herbicide (Figure 5). Transcriptome analysis revealed that 4139 differentially expressed genes were at a log_2_ > |2| level (Figure 5A), and the Kyoto Encyclopaedia of Genes and Genomes (KEGG) enrichment results showed that the differential expression pathways were concentrated in the carbon and nitrogen metabolism, photosynthesis, and secondary metabolism pathways, which overlapped to some extent with the inference from the fluorescence results. However, the KEGG enrichment results at the log_2_ > |5| level shifted towards endoplasmic reticulum (ER) stress, lipid metabolism, and other pathways that favour membrane structure damage (Figure 5B). To validate the ER stress pathway, Gene Set Enrichment Analysis (GESA) analysis was performed (Appendix A) and showed that the pathway was enriched in cell death pathways, such as ER stress. Programmed necrosis in plants can occur through the dysregulation of ER stress and chloroplast homeostasis pathways. The enriched ER stress genes mainly consisted of heat stress proteins (Figure 5F). To clarify the key nodes of ER stress, we performed key driver analysis (KDA) on the genes in the top 500 of the log2 difference and the ER stress pathway in GESA. We obtained a total of 14 key driver genes with clear annotations, from which, 7 genes were selected for qPCR validation. The transcriptome and the qPCR results are consistent with each other, and the validation results are reliable (Figure 5C). Treatment with imazethapyr caused changes in ER stress and related pathways at the transcriptional level in leaf cells. Additionally, photosynthesis, carbon and nitrogen metabolism, and DNA synthesis were blocked. The KEGG network analysis of the top 500 differentially expressed genes showed an MAPK cascade response pathway associated with the ER stress pathway through the pathogen immune stress pathway (Figure 5D). However, the KEGG mapper only revealed the presence of the calcium pathway and PR1 protein expression for all MAPK genes and the KEGG network associated with plant–pathogen genes (Appendix A), with no other pathogen receptor involvement. Therefore, it can be inferred that the development of necrotic spots is not associated with biotic stress and resembles a hypersensitive response. The identification of ER stress-related genes enriched by log_2_ > |5| revealed two genes involved in immune death-related genes, namely *SKP1* and *CALlR,* which were consistent with the qPCR results.

The KEGG enrichment analysis showed that the enriched ER stress pathway contained a significant number of heat shock proteins, known to be more sensitive to heat. The heat shock transcription factors were then subjected to clustered heat map analysis, revealing the presence of *HSFA2* and *HSFB2* family transcription factors with increasing expression. The *HsfB2c* gene was selected for qPCR verification, and the results were consistent with the clustering results (Figure 5C,E). It is speculated that a high temperature may also affect the occurrence of necrotic spots.

### 2.4. Location Detection of Intracellular Necrosis

The chloroplast status of the necrotic surrounding cells (Figure 6A) indicates that HRT1 caused a significant accumulation of temporary starch granules and cloudy vacuole fluid and blocked cell division after treatment with imazethapyr. Further observation of the treated group (Figure 6B,C) revealed that despite the significant synthesis of starch granules, the chloroplast morphology remained relatively regular, with an intact and clear membrane and thylakoid structures. Additionally, mitochondrial depolarisation was low, while fragmented ER organisation and liposomes were observed within the vacuoles.

The transcriptome was analysed to classify and enrich the cellular composition based on Gene Ontology (GO) (Figure 6D,E). The analysis revealed that the proportion of differentially expressed genes that act on chloroplasts and mitochondria, the sources of reactive oxygen species (ROS), was low. Additionally, the GO enrichment results showed that differential expression was concentrated on the membrane structures in the cytoplasm. Cell necrosis is frequently caused by a sudden burst of ROS and a continuous dysregulation of homeostasis. ROS preferentially attack the tissues surrounding the site of necrosis. Chloroplasts, the largest site of ROS production in plant cells, preferentially disintegrated the membrane structure of chloroplasts and mitochondria when the necrosis in HRT1 was due to chloroplast-associated defects. However, this speculation was not supported by the electron microscopy results. The transcriptome results, combined with the electron microscopy results, suggest that the critical sites for the development of intracellular necrosis should not be on the chloroplasts or mitochondria.

### 2.5. Response of Necrotic Phenotype to Temperature and Light

This study investigated the impact of light and temperature on herbicide treatment-induced leaf necrosis in weedy rice. One-way and two-way temperature–light experiments were conducted under ambient temperature and shade–no shade conditions. The results showed significant changes in leaf area and Fv/Fm (Figure 7B(1–3),C) in the single-factor experiments of temperature and light. The decrease in Fv/Fm was greater in both the intense-light and high-temperature groups than in the control group. This suggests that both intense light and high temperatures promote the production of necrotic spots. The temperature–light experiments (Figure 7A,B(4) and Appendix A) showed that the high-temperature treatment had a greater effect on necrotic spots than intense light.

The transcriptome was analysed to classify and enrich the cellular composition based on GO (Figure 6D,E). The analysis revealed that the proportion of differentially expressed genes that act on chloroplasts and mitochondria, the sources of ROS, was low. Additionally, the GO enrichment results showed that the differential expression was concentrated on the membrane structures in the cytoplasm. Cell necrosis is frequently caused by a sudden burst of ROS and a continuous dysregulation of homeostasis. ROS preferentially attack the tissues surrounding the site of necrosis. Chloroplasts, the largest site of ROS production in plant cells, preferentially disintegrated the membrane structure of chloroplasts and mitochondria when the necrosis in HRT1 was due to chloroplast-associated defects. However, this speculation was not supported by the electron microscopy results. The transcriptome results, combined with the electron microscopy results, suggest that the critical sites for the development of intracellular necrosis should not be on chloroplasts and mitochondria.

## 3. Discussion

### 3.1. Identifying a New Phenotype of Necrotic Spot in Weedy Rice Responding to IMI Herbicides

This study reported the rapid response of weedy rice to IMI herbicides, resulting in a new phenotype of leaf necrotic spots. These findings provide new insights for analysing the efficacy of rice IMI herbicides and the control of weedy rice.

The biggest challenge facing IMI herbicide-resistant rice varieties is the frequent occurrence of resistant weedy rice and the overall increase in herbicide resistance. A survey of local farmers in Malaysia has shown that some weedy rice survives during the season, and resistant weedy rice emerges after five to six seasons of Clearfield^®^ rice [16]. It is now accepted that such resistant weedy rice is mainly due to the drift of resistance genes [17]; therefore, suppressing the weedy rice population and gene flow is critical and essential [18]. IWMS is currently the most appropriate management model to control weedy rice and extend the technical life of the herbicide resistance system while taking benefits into account [19].

In this system, chemical methods are required to reduce weeds without reducing efficiency and adapt to local situations [20]. IMI herbicides have calm and slow overall efficacy but high residual amounts of weeds [15]. It is difficult to ensure a suitable concentration in cultural practice. Referring to Section 2.2.2, a difference in efficacy between the technical product and formulation. Although the growth of HRT1 was inhibited under 1- and 2-fold treatment of the technical product, no obvious necrotic spots were formed. After 7 days, the photosynthetic fluorescence state had basically recovered to the same level as the control. According to a study [21], weedy rice has a stronger growth rate and competitive ability than cultivated rice. Poor efficacy may not allow cultivated rice to form an absolute growth advantage over more tolerant weedy rice in just one week. Meanwhile, IMI herbicides are inhibitors of ALS, which also belong to the class of amino acid inhibitors [15]. Casey and Xu demonstrated that applying BCAA in the early and middle stages could reverse or substantially reduce the effect of an ALS inhibitor [22,23]. Nitrogen fertilisers or growth regulators are often applied with herbicides at nearly the same time. These substances also act as antidotes to IMI [24]. Thus, comparing the conventional type and the leaf necrosis spot type, which occurs rapidly and cannot recover, can support cultural practices to match.

### 3.2. ER Stress Possibly Causes Necrotic Spots under Imazethapyr Treatment

This study provides preliminary evidence for a novel mechanism by which imazethapyr induces ER stress, leading to the generation of necrotic spots in weedy rice. The physiological mechanisms of the ALS inhibitor classes on plants have been largely elucidated, the most notable of which is their effect on plant nitrogen metabolism. In the short term, there is a large accumulation of free amino acids, especially BCAA, and other amino acids are rapidly degraded [25], which is in agreement with the transcriptome results of this experiment. Zabalza et al. found with N_15_ that imidazole ethanolic acid application leads to a decrease in root vigour as well as the impairment of nitrogen uptake accompanied by stomatal closure in soybean [25]. Correspondingly, Zabalza found that imazethapyr application leads to the accumulation of transient starch in chloroplasts, resulting in the starch grain accumulation effect shown in this paper [26]. Qian’s results were consistent with the above [27]. The starch deposition was attributed to the decrease in root vigour and stomatal conductance and, consequently, the reduction in the overall intensity of carbon and nitrogen cycling. At the cellular level, the effects of ALS enzymes on chloroplasts are more pronounced due to their expression in chloroplasts. In addition to causing the accumulation of starch granules, it can have destructive effects, such as chlorophyll degradation, thylakoid membrane breakdown, and reduced photosynthetic efficiency as it accumulates over time, ultimately leading to the disintegration of chloroplasts [28]. At the same time, it causes mitochondria to undergo anaerobic respiration, and AOX enzyme activity increases significantly [29]. The accumulation of multiple deleterious effects leads to a gradual increase in cellular ROS, which ultimately results in autophagic apoptosis. This feature is significantly expressed in pollen, and taking advantage of this feature, ALS inhibitors are often used as anthericides in cruciferous crops [28,30]. Microspores and tapetum cells show distinct autophagic vacuole structures after treatment. The ITS and leaf chloroplasts are deformed, and the transcriptome is consistent with the cytological results [31,32], which overlap, to some extent, with the cytology and transcriptome of the present experiment. However, ER stress-related pathways have not been mentioned in other studies.

The necrotic spot phenotype in this experiment showed high similarity to environmentally induced rice disease-like spots, and the possibility of biotic stress was largely ruled out by microscopic examination and transcriptome analysis. Thus, using the mechanisms of development of this class of spots as a reference, most of them are caused by a mutation in a gene that leads to the active and susceptible disruption of a certain part of the cell or a decrease in resistance to homeostatic perturbations, and all of them ultimately lead to a burst of ROS that triggers a hypersensitive response, leading to programmed death [33,34].

Since the transcriptome log_2_ > |2| and log_2_ > |5| results indicated chloroplasts and ER, respectively, the cause of intracellular necrosis was not clear. Drawing on the male-killing mechanism of ALS inhibitors, it was tentatively hypothesised that defective chloroplasts might generate excessive ROS, triggering a hypersensitivity reaction that selectively damages the chloroplast structure [35]. However, electron microscopy and GO structural analysis revealed that chloroplasts and mitochondria did not undergo significant disintegration, and mitochondrial stress is also a major type of necrotic pathway in plants [36]. Therefore, the possibility of chloroplast- and mitochondria-related changes as the main cause of necrotic death was ruled out. The two highly differentially expressed genes were concentrated in ER stress, lipid metabolism, and secondary metabolism, and electron microscopy results also showed the presence of degraded and swollen ER structures in the vesicles. Meanwhile, the light–temperature verification test showed that temperature treatment with a better correlation with heat-stimulated proteins had a greater effect on necrosis [37]. Therefore, it was finally concluded that the key site of intracellular necrosis was in the ER and was related to the state of the ER itself, which triggered a similar hypersensitivity mechanism, leading to the generation of necrotic spots on the leaves.

Based on the experimental results and literature investigation, there are two possible mechanisms. First, the ER of this rapidly necrotic type of weedy rice is less resistant to non-homeostasis compared to non-rapid necrotrophic weedy rice, and the dysregulation of amino acid metabolism after imazethapyr treatment leads to an increase in the number of unfolded proteins on the ER [38]. This is also reflected in the increased transcript levels of heat stress proteins in response to protein homeostasis, which triggers unprogrammed death [39]. However, only the transcript level of the ER receptor gene *IRE1* was found to be up-regulated in the ER stress pathway marker gene assay in plants, which may have been due to the early sampling time. *IRE1* may also be involved in necroptosis independent of ER stress in animals, but this pathway has not been detected in plants [40,41]. The next step should be to examine the ER status before and after imazethapyr treatment and at different times of the day using sectioning and qPCR. In the meantime, although methomyl fumarate was also verified in the present study, and the preliminary demonstration showed that IMIs can induce necroptosis, other herbicide types have not been validated. Second, KEGG network analysis and mapping showed the existence of a plant defence pathway triggered by calcium ions and some related heat stress proteins, and transcript levels of immune surveillance genes were up-regulated, but whether this was induced as a cause of the event or by dysregulation of the ER could not be determined [42]. Further tests for bacterial disease resistance and calcium homeostasis-regulated genes should be carried out to search for mutant genes or common pathways in weedy rice.

### 3.3. Research Prospects of New Phenotypes of Necrotic Plaques in Herbicide Reaction and Production

In this study, necrotic spots or mixed types were found in half of the test population of weedy rice, and the necrotic type was present in most of the population, indicating that there was no group tendency for this type. The experiment clarified that both intense light and high temperature can effectively induce necrotic development through the experiments of one- and two-factor photosynthetic fluorescence and phenotyping tests of light and temperature, but whether there is a synergistic effect between the two needs to be investigated by further relevant experiments. We found many heat-expressed proteins involved in protein homeostasis and immune response through transcriptional analysis, and heat-expressed proteins can be regulated by light and heat stress [43,44]. Both stressors can increase the adversity stress of cells, and it is worthwhile to explore whether such environmental stressors are involved in key pathways of necroptosis. Most interestingly, imazethapyr is an endophytic herbicide, but its application to this kind of weedy rice shows the injury phenotype commonly seen with contact herbicides. Is this caused by transport barriers or environmental factors? Understanding the mechanism may provide new ideas in terms of herbicide efficacy.

## 4. Materials and Methods

### 4.1. Plant Materials

A total of 113 weedy rice samples were collected from 16 countries worldwide (China, South Korea, Japan, Myanmar, Laos, Cambodia, Vietnam, Philippines, India, Nepal, Bhutan, Sri Lanka, Brazil, and the USA). Namweonaengmi 2, coded as HRT1 (herbicide response type 1), has a responding phenotype of typical necrotic spots and comes from South Korea. SZ43, coded as HRT2 (herbicide response type 2), with a responding phenotype of yellowing and wilting, comes from Jiangsu Province, which plants an IMI-R rice variety. Jingeng 818, an IMI-R variety, is planted in Jiangsu Province in China.

### 4.2. Statistics of Response Types to Imazethapyr in the Weedy Rice Population

The population was planted with six plants in each hill after germination and sprouting, and the whole plant was treated and measured during the two-leaf stage with a water control and imazethapyr treatment (Alfalfa Clear, 150 g a.i. hm^−1^, imazethapyr aqueous solution bought from Shandong Xianda Agrochemical Co., Ltd., Weifang, Shandong, China). Alfalfa Clear (225 μL) was dissolved in 45 mL of sterile water and sprayed evenly with a 0.8 L hand-held sprayer (Ichishita brand, Ichishita Holding Co., Ltd., Taizhou, China) at a spray pressure of 0.2 Mpa. The phenotypes of herbicide damage were counted once a week for 2 weeks, and the standard of herbicide damage was measured following Haider et al.’s method [45].

### 4.3. Identification of Necrosis Spot

#### 4.3.1. Phenotype Observation

Single pots of HRT1 and HRT2 with good imazethapyr tolerance were selected and grown in the greenhouse to the three-leaf stage, with HRT2 planted 7 days earlier and HRT1 and HRT2 treated with the formulation at the field-recommended dose of 112.5 g a.i. hm^−1^ and photographed for comparison when both phenotypes were evident.

#### 4.3.2. Microscopic Examination

First, leaves with necrotic spots were removed, and the necrotic spots were observed using a Zeiss Smartzoom 5 super depth field microscope (Oberkochen, Baden-Württemberg, Germany). Then, the freshly spotted parts were collected, fixed in 2% glutaraldehyde, and sent to Sevier (Wuhan, Hubei, China) for transmission electron microscopy. Lastly, 1 cm^2^ of the diseased and healthy junctions were cut to make temporary sections, and a Leica Axiostar plus fluorescence microscope (Leica microsystem, Wetzlar, Hessian, Germany) was used to observe the phenomenon of fungal spores in bright field and to take photographs.

### 4.4. Relationship between Imazethapyr and Necrosis Spot

#### 4.4.1. Formulation Smear Test

Planting conditions were the same as in Section 4.3.1. When the seedlings were in the three-leaf stage, a brush was moistened with 75 g a.i. hm^−1^ of the formulation and applied to the middle and upper part of the second leaf blade, taking care not to let the liquid run down to the underside of the leaf. The phenotypic changes were observed, and photographs were taken.

#### 4.4.2. Efficacy Comparison between the Technical Product and the Formulation

The imazethapyr technical product (TC) group (purchased from Shanghai McLean Biochemical Science and Technology Co., Ltd., Shanghai, China) was tested at concentrations of 0, 37.5, 75, 150, 300, and 600 g a.i. hm^−1^, which were the control and 1, 2, 4, and 8 times the recommended dose, respectively. The formulation of the AS (aqueous solution) concentration was 75 gai hm^−1^. Each group of 50 HRT1 plants were planted in 30 cm × 40 cm pots, with three replicates, and grown to the two- to three-leaf stage. The degree of necrosis was measured using the following scale: 0, no necrosis; 1, appearance of necrotic spots; 2, necrotic spots aggravated or aggravated to form an irregular streak. Statistics were calculated. Meanwhile, another 6 single seedlings were planted for each group, and photosynthetic fluorescence observation was carried out 1 day before spraying and 3, 5, and 7 days after spraying using PlantExplorer, a multifunctional plant photosynthetic phenotypic imaging and measurement system (PhenoVation, Wageningen, Gelderland, The Netherlands). The standard value of Fv/Fm was 0.83, and the relationship between IMIs and photosynthetic fluorescence has been demonstrated in previous studies [46,47].

#### 4.4.3. Resistant Variety Verification

Jingeng 818 (*O. sativa* ssp. *japonica*, Jiangsu IMI-resistant variety) was planted in the three-leaf stage under the same conditions as in Section 4.3.1, and five gradient spraying treatments (0, 37.5, 75, 150, and 300 g a.i. hm^−1^) were applied to observe the phenotypic changes.

### 4.5. Transcriptional Level Analysis

#### 4.5.1. Transcriptomics

HRT1 was planted in the three-leaf–one-heart stage, and control and herbicide treatment groups were established. The imazethapyr spray treatment was applied when the new leaves were just emerging, and the third whole leaf was collected when the new leaves were wilted. Five leaves from each group were collected as one replication, and a total of three replications were sent to Shenzhen Huada Gene Science and Technology Co., Ltd. (Shenzhen, Guangdong, China) for transcriptome sequencing and analysis using their biosignature platform, BGI Ltd. (http://report.bgi.com). BioProject: PRJNA1085844.

#### 4.5.2. Q-rtPCR

Seven predicted key driver genes and one HSF gene were verified by qPCR, and the remaining RNA from the transcriptome was used for reverse transcription using a reverse transcription kit (HiFiScript cDNA Synthesis Kit, Kangwei Century Co., Ltd., Beijing, China). Reverse transcription was performed, and primers were designed by NCBI and synthesised by Shanghai Jierui Bioengineering Co., Ltd. (Shanghai, China). qPCR experiments were performed using the Power SYBR™ Real-Time PCR Kit and Green PCR Premix (Thermo, Waltham, MA, USA). Primers are listed in Appendix A.

### 4.6. Cytological Verification

Three-leaf-stage HRT1 plants were organised into two groups, water control and imazethapyr treatment, which were treated and left until necrotic spots appeared. Samples of the diseased and healthy junction and the normal group with the same treatments as in Section 4.3.2 were collected for transmission electron microscopy observation.

### 4.7. Influence of Temperature and Light on Necrotic Spots

#### 4.7.1. Light Power

The treatment group, covered light (CL), and the control group, uncovered light (UCL), were established for phenotypic and fluorescence observation. Plants were placed in the greenhouse outside the roof, and planting conditions followed those in Section 4.3.1. There was a comparative check after spraying. Photosynthetic fluorescence observation was performed as described in Section 4.3.2. The 5 shading treatments were established using a shade net as follows: shading effect of about 50%, outdoor light intensity 193 × 1000 lx, roof and inside-greenhouse light intensity of 97 × 1000 lx, and shading intensity of 51 × 1000 lx. Fluorescence was observed 1 day before and 6 days after spraying.

#### 4.7.2. Temperature

The common-temperature (CT) and high-temperature (HT) treatments were set up to be consistent with the experiments in Section 4.7.1, with the common-temperature group being placed outdoors and the high-temperature group being placed inside the greenhouse, where high temperatures of up to 30 °C or more could be reached on a sunny day. The attached figure shows the specific temperatures (Appendix A).

#### 4.7.3. Temperature and Light Power Co-Treatment

HRT1 was planted during the two-leaf stage; selected seedlings with uniform growth were transferred to individual pots, and when the three-leaf stage was reached, herbicide treatment was applied at a concentration of 75 g a.i. hm^−1^, according to the two-factor temperature and light experiment, with light and temperature in accordance with Section 4.7.1 and Section 4.7.2. Fluorescence was observed 1 day before and 3 and 6 days after spraying.

## 5. Conclusions

In this experiment, the phenotypic characteristics and occurrence of a novel phenomenon showed that imazethapyr could rapidly generate necrotic spots in some weedy rice. Necrotic spots were identified as a physiological type of hypersensitive programmed death and were strongly correlated with imazethapyr and an environment characterised by high temperature and intense illumination. When comparing the efficacy of the formulation and the technical product, the involvement of the formulation enhanced the efficacy by about four times. Still, it was not effective in killing the more tolerant weedy rice when the dosage was insufficient. It is recommended that the application be made under sunny weather conditions to maximise the efficacy of the herbicide. When further investigating the occurrence of necrotic spots, in addition to known chloroplast and mitochondrial damage, imazethapyr also caused ER stress and partial hypersensitivity pathway expression in weedy rice leaf cells. In conclusion, the results of this study enrich the patterns of response to IMI herbicides in weedy rice and provide new ideas for improving the efficacy of herbicides and the management of weedy rice.

## Figures and Tables

**Figure 1 plants-13-01218-f001:**
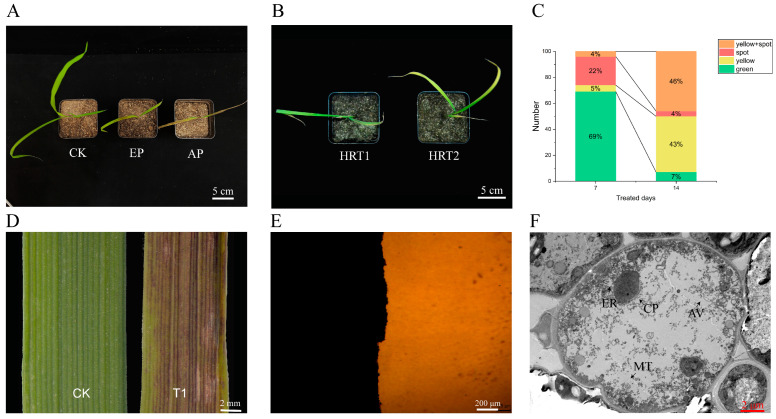
Investigation and identification of the necrosis phenotype. (**A**) Necrotic process, CK: water treated for control, EP: early phase, AP: later phase. (**B**) Necrotic and losing green phenotype. (**C**) Investigation of the phenotype of imazethapyr induced by natural population toxicity in weedy rice. Green: keep green, yellow: lose green, spot: necrotic spot. (**D**) Leaf surface of normal growth and necrosis. (**E**) Necrosis leaf microscopic examination. (**F**) Necrotic spots, magnification 3000 times by transmission electron microscope. CP: chloroplast, MT: mitochondria, AV: autophagic vacuole, ER: endoplasmic reticulum.

**Figure 2 plants-13-01218-f002:**
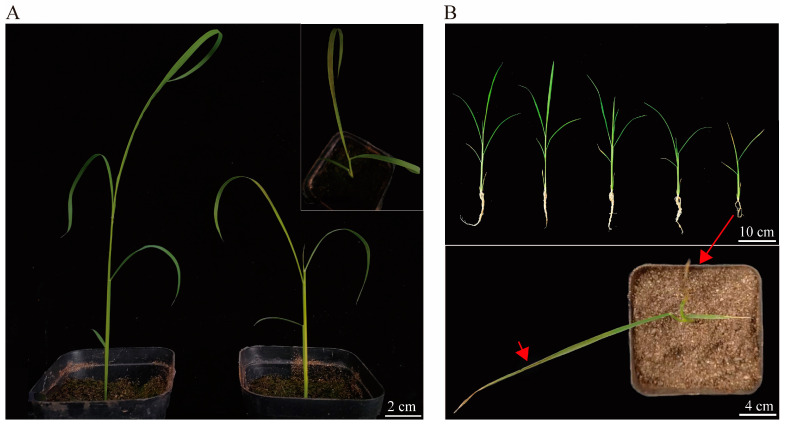
Relationship between imazethapyr and necrotic plaques. (**A**) Smear test with 75 g a.i. ha^−1^. (**B**) Jingeng818 IMI gradient processing: 0, 37.5, 75, 150, and 300 g a.i. ha^−1^ from left to right.

**Figure 3 plants-13-01218-f003:**
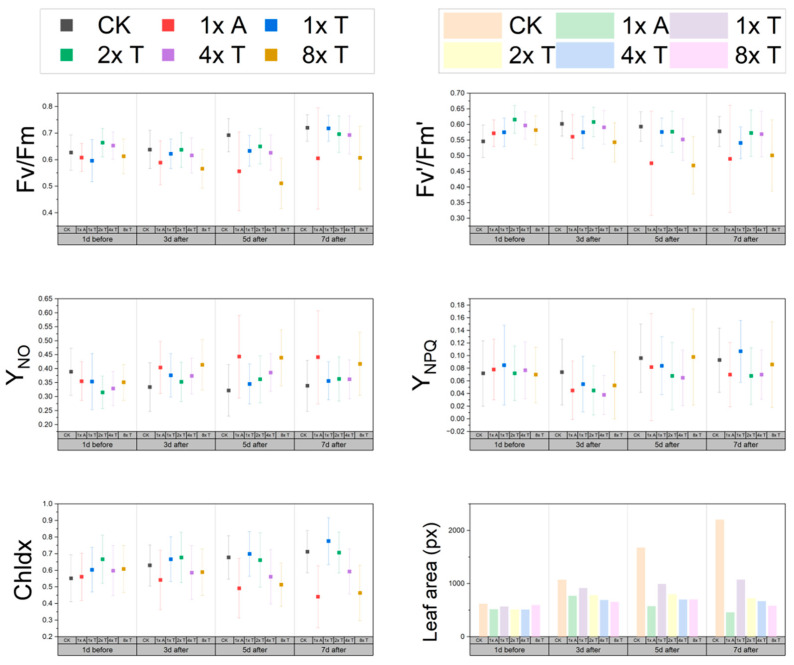
Comparison of the efficiency of the technical product and the formulation by photosynthetic fluorescence parameters. The trend of 1x A and 8x T is basically the same, and the standard deviation is increasing. Fv/Fm: maximum photosynthetic efficiency, Fv’/Fm’: actual photosynthetic efficiency, Chldx: the chlorophyll index, Y_NO_: quantum yield of the nonregulated energy dissipation, Y_NPQ_: photo-quantum yield of the regulated energy dissipation, Leaf area: average leaf area of six plants.

**Figure 4 plants-13-01218-f004:**
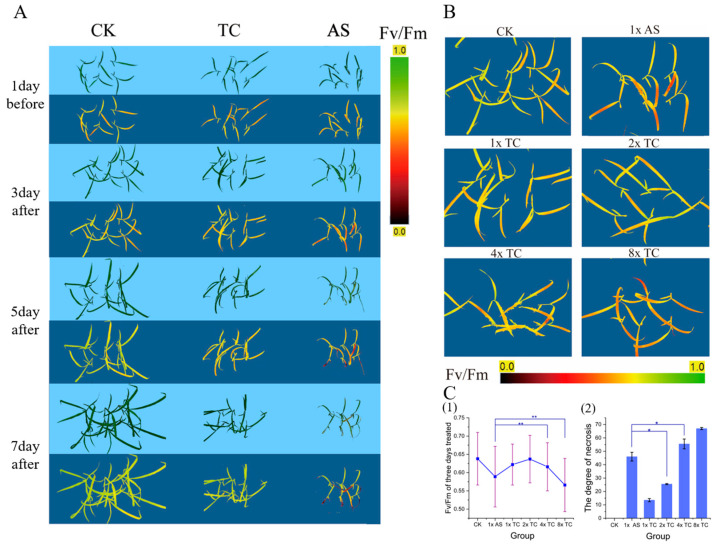
Occurrence pattern and fluorescence imaging of necrotic spots. (**A**) Fv/Fm and phenotype variation in 1x A, 1x T, and control. (**B**) Fv/Fm comparison of the pure chemical and the product on the 3rd day. (**C**) The change of necrotic spots degree between different treatments, (**C1**) Chart showing the Fv/Fm on the 3rd day; (**C2**) necrosis on the 7th day between the pure chemical and product. * and ** showed significant (*p* < 0.05) and extremely significant (*p* < 0.01), respectively.

**Figure 5 plants-13-01218-f005:**
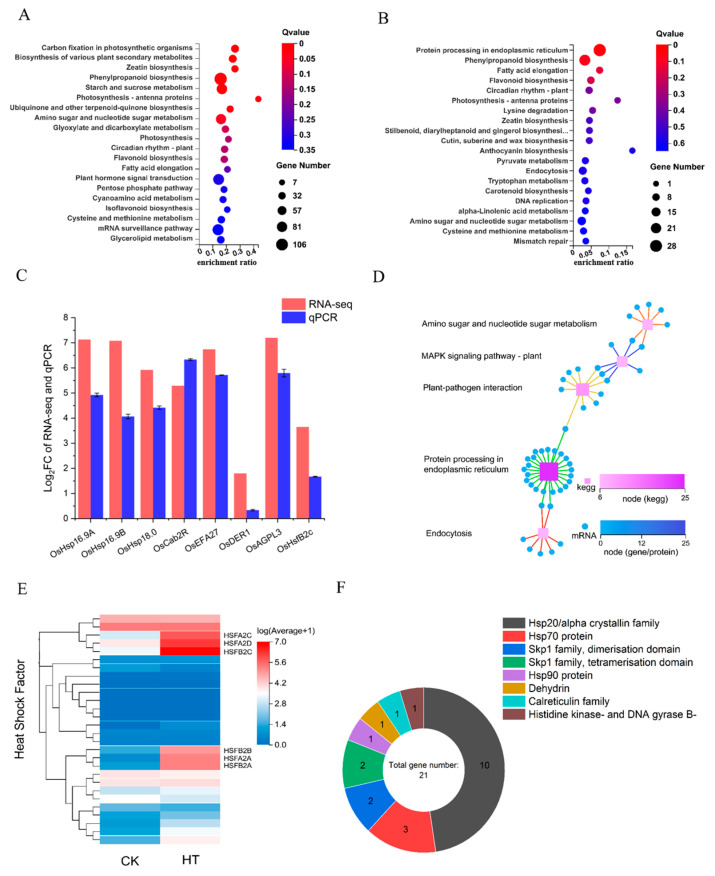
Transcriptional analysis of necrotic materials and qPCR validation. (**A**) KEGG enrichment analysis of log_2_ > |2|, focusing on photosynthesis, carbon and nitrogen metabolism, and secondary metabolism. (**B**) KEGG enrichment analysis of log_2_ > |5|. At this level, programmed cell death pathways, such as ER stress, lipid metabolism, and secondary metabolism, are significantly enriched. (**C**) Comparison of qPCR and RNA-seq selected by KDA analysis. (**D**) KEGG network enrichment analysis, selecting the top 500 of log_2_ difference. (**E**) Clustering heat map of HSF. HT: herbicide treatment. (**F**) Classification of ER stress pathway enriched from (**B**).

**Figure 6 plants-13-01218-f006:**
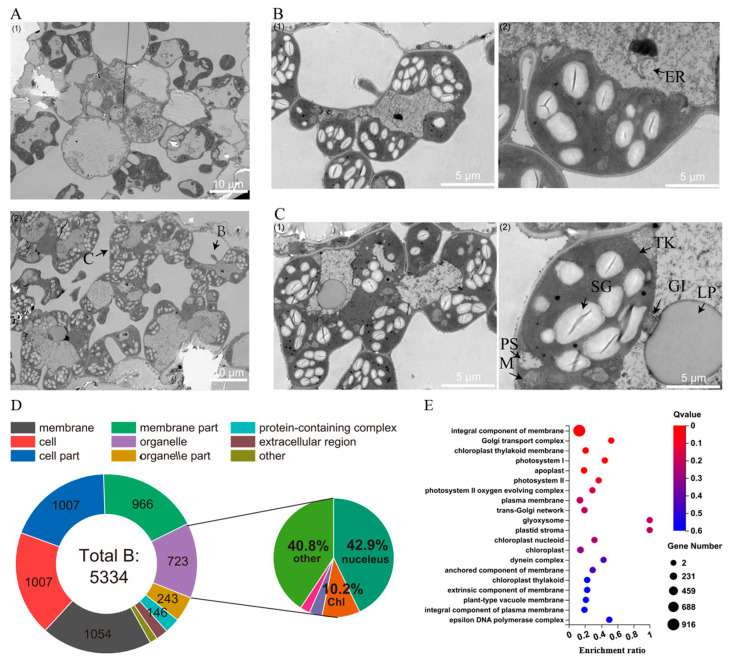
Location detection of intracellular necrosis. (**A**) TEM between CK and necrotic margin: (**1**) is CK, (**2**) is necrotic margin. (**B**) Detailed part1 (1) cellular level; (2) Organelle level, (**C**) follow this, ER: endoplasmic reticulum. (**C**) Detailed part 2, TK: thylakoid, SG: starch grain, PS: phagosome, M: mitochondria, GI: Golgi, LP: lipidosome. (**D**) GO Cellular Component analysis of log_2_ > |2|; in the small circle, the purple part belongs to mitochondria, 2.9%. (**E**) GO enrichment analysis of organelles enriched from (**D**).

**Figure 7 plants-13-01218-f007:**
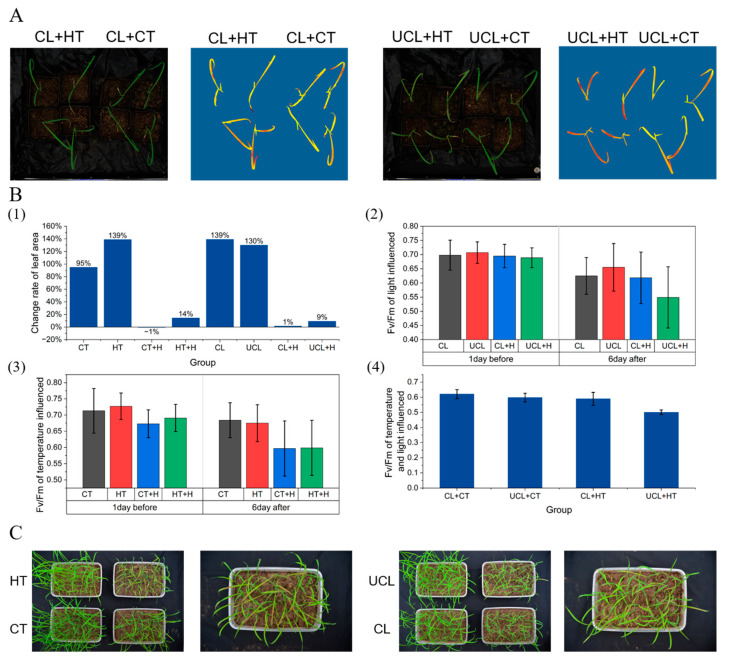
Effect of intense light and high temperature on necrotic spots. (**A**) Fv/Fm on the 6th day of imazethapyr treatment with the temperature–light influence; CL: cover light, CT: common temperature, UCL: uncover light, HT: high temperature. (**B**) Effect of imazethapyr on the light, temperature, and light–temperature factor. (**1**) Leaf area changing rate of single light and temperature, H: herbicide. (**2**) Fv/Fm of light–herbicide. (**3**) Fv/Fm of temperature–herbicide. (**4**) Chart showing (**A**), (**C**) phenotype of temperature and light treatment.

## Data Availability

Data are contained within the article.

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
