# Peer review of "Phenotypic Characteristics and Occurrence Basis of Leaf Necrotic Spots in Response of Weedy Rice to Imazethapyr"

_plants, 2024, doi:10.3390/plants13091218_

Round 1
Reviewer 1 Report
Comments and Suggestions for Authors
The manuscript “Characteristics and Occurrence Basis of Leaf Necrotic Spots in Weedy Rice Response to Imazethapyr” demonstrated that imazethapyr can rapidly induce necrotic spots in certain weedy rice varieties, a phenomenon linked to high temperature and intense light. The formulation of imazethapyr significantly enhanced efficacy, but insufficient dosage failed to eradicate more tolerant strains. It was also concluded that application under sunny conditions is recommended for optimal results. The manuscript was professionally produced, and it merits publication in the Plants journal following minor revisions. Suggestions for the authors to consider are outlined below.
Lines 30-32: Review: Weedy rice (Oryza sativa f. spontanea) is a term used to describe rice with weedy characteristics that is typically found in rice fields and its surrounding areas and is an intermediate species between wild and cultivated rice.
Suggestion: Weedy rice (Oryza sativa f. spontanea) is a term used to describe rice with weedy characteristics. It is typically found in rice fields and their surrounding areas, serving as an intermediate species between wild and cultivated rice.
Lines 104-107: However, by observing the leaf surface and microscopic examination (Figure 1D and 1E), this necrotic spot had no obvious bacterial pus production and no bacterial spraying phenomenon, which excluded the possibility of bacteria being the main cause of occurrence.
Suggestion: However, by observing the leaf surface and microscopic examination (Figure 1D and 1E), this necrotic spot did not show obvious bacterial production of pus and no phenomenon of bacterial proliferation, which excluded the possibility of bacteria being the main cause of occurrence.
Comments on the Quality of English LanguageThe manuscript was professionally produced, and it merits publication in the Plants journal following minor revisions.
Author Response
Dear reviewer
Thank you very much for taking the time to review this manuscript. Please find the detailed responses below and the corresponding revisions/corrections highlighted/in track changes in the re-submitted files.
Comments 1:
Lines 30-32: Review: Weedy rice (Oryza sativa f. spontanea) is a term used to describe rice with weedy characteristics that is typically found in rice fields and its surrounding areas and is an intermediate species between wild and cultivated rice.
Suggestion: Weedy rice (Oryza sativa f. spontanea) is a term used to describe rice with weedy characteristics. It is typically found in rice fields and their surrounding areas, serving as an intermediate species between wild and cultivated rice.
Response 1:
Thank you for pointing this out. We agree with this comment. Therefore, we have revised the sentence according to your suggestion.
“Weedy rice (Oryza sativa f. spontanea) is a term used to describe rice with weedy characteristics. It is typically found in rice fields and their surrounding areas, serving as an intermediate species between wild and cultivated rice.” at the lines 30-32
Comments 2:
Lines 104-107: However, by observing the leaf surface and microscopic examination (Figure 1D and 1E), this necrotic spot had no obvious bacterial pus production and no bacterial spraying phenomenon, which excluded the possibility of bacteria being the main cause of occurrence.
Suggestion: However, by observing the leaf surface and microscopic examination (Figure 1D and 1E), this necrotic spot did not show obvious bacterial production of pus and no phenomenon of bacterial proliferation, which excluded the possibility of bacteria being the main cause of occurrence.
Response 2:
Agree, we have revised the sentence.
“This necrotic spot has some similarity to the spots caused by the brown spot pathogen. However, by observing the leaf surface and microscopic examination (Figure 1D and 1E), this necrotic spot did not show obvious bacterial production of pus and no phenomenon of bacterial proliferation, which excluded the possibility of bacteria being the main cause of occurrence.” At the lines 101-107

Reviewer 2 Report
Comments and Suggestions for Authors
Dear Authors,
Overall, the work was well developed in the experimental aspects, with relevant findings. The experimental results provide valuable insights into the process and basis of imazethapyr in inducing the rapid generation of necrotic spots in rice leaves. Additionally, the findings contribute to a better understanding of the mechanism of response to imidazolinone herbicides and the control of weedy rice, which can help in developing effective strategies for weed management in rice production.
I think that, perhaps, a further investigation could be conducted to validate the findings of laboratory experiments and transcriptome analysis, considering the complexity and variability of real-world agricultural conditions.
I have some suggestions:
Abstract
Line 12: Change “difficult” by “challenging”
Line 21: Change “during the occurrence of necrosis” by “during necrosis”
Line 23: Change “strong light” by “intense light”
Introduction
Line 31: Exclude “that is”
Line 34: “…competitive advantages, disease resistance, and reproductive…”
Lines 37-38: Change “led to an increase in” by “increased…”
Line 64: Change “some of the weedy” by “some weedy”
Results
Line 68 and others: The word "obvious" can be replaced with synonyms such as "apparent", "evident", "prominent", or others. Please review the manuscript.
Line 205: Change “were basically consistent” by “were consistent”
Line 220: “shock proteins, which are known to be more sensitive to heat.” By “shock proteins, known to be more sensitive to heat.”
Discussion
Line 318: “Casey and Xu demonstrated that applying BCAA…”
Line 407: “We found a large number of heat-expressed” by “We found many heat-expressed”
Line 426: “gemination” by “germination”
Conclusions
Line 526: “imazethapyr could induce the rapid generation of necrotic spots “ by “imazethapyr could rapidly generate necrotic spots”
Line 531: “four times, but it was not effective” by “four times. Still, it was not effective”
Appendix is OK.
Figures
Figures 1C, 3D, and 4C could be colored to enhance visibility.
I believe that Figure 3A, which displays photosynthetic fluorescence parameters and leaf area, is not visually clear and could be improved. It is quite difficult to interpret due to its unclear appearance. Perhaps it would be better to present the data in a separate figure or improve its visual design. Alternatively, it could be more intense lines and symbols.
Figures 6 B (1 and 4) could be colorful, while 2 and 3 could be more intense lines and symbols.
Comments on the Quality of English LanguageThe manuscript is well written, requiring minor proofreading.
However, I suggest that some figures be improved, as they are not visually acceptable.
Author Response
Dear reviewer
Thank you very much for taking the time to review this manuscript. Please find the detailed responses below and the corresponding revisions/corrections highlighted/in track changes in the re-submitted files.
Comments 1:
I think that, perhaps, a further investigation could be conducted to validate the findings of laboratory experiments and transcriptome analysis, considering the complexity and variability of real-world agricultural conditions.
Response 1:
We appreciated for your suggestion. It is better to be conducted the further investigation, but we have no the suitable materials with similar genetic background at present.
Comments 2:
Abstract
Line 12: Change “difficult” by “challenging”
Response: “Weedy rice is the most challenging weed species to remove in rice production.”
Line 21: Change “during the occurrence of necrosis” by “during necrosis”
Response: “the lipid metabolism membrane structure damage pathway during necrosis,”
Line 23: Change “strong light” by “intense light”
Response: “showed that high temperature and intense light could promote the appearance of necrotic spots”. We agree the strong is quite appropriate, so we replace all in this article.
Introduction
Line 31: Exclude “that is”
Response: “Weedy rice (Oryza sativa f. spontanea) is a term used to describe rice with weedy characteristics. It is typically found in rice fields and their surrounding areas, serving as an intermediate species between wild and cultivated rice.”
Line 34: “…competitive advantages, disease resistance, and reproductive…”
Response: “As wild relatives, they possess stronger competitive advantages, disease resistance and reproductive characteristics,”
Lines 37-38: Change “led to an increase in” by “increased…”
Response: “…rice direct seeding areas increased in the incidence…”
Line 64: Change “some of the weedy” by “some weedy”
Response: “We observed that some weedy rice showed necrotic spots…”, please confirm line 62-63.
Line 68 and others: The word "obvious" can be replaced with synonyms such as "apparent", "evident", "prominent", or others. Please review the manuscript.
Response: “their drawbacks are also apparent.”
Results
Line 205: Change “were basically consistent” by “were consistent”
Response: “the qPCR results consist with each other…”, please confirm line 207.
Line 220: “shock proteins, which are known to be more sensitive to heat.” By “shock proteins, known to be more sensitive to heat.”
Response: “The KEGG enrichment analysis showed that the enriched ER stress pathway contained a significant number of heat shock proteins, known to be more sensitive to heat.”, please confirm line 222.
Discussion
Line 318: “Casey and Xu demonstrated that applying BCAA…”
Response: “Casey and Xu demonstrated that applying BCAA in the early and middle stages”
Line 407: “We found a large number of heat-expressed” by “We found many heat-expressed”
Response: “We found many heat-expressed proteins involved in protein homeostasis…”
Material
Line 426: “gemination” by “germination”
Response: “The population was planted with six plants in each hill after germination and sprouting,”
Conclusions
Line 526: “imazethapyr could induce the rapid generation of necrotic spots “ by “imazethapyr could rapidly generate necrotic spots”
Response: “In this experiment, the phenotypic characteristics and occurrence of a novel phenomenon showed that imazethapyr could rapidly generate necrotic spots in some weedy rice.”
Line 531: “four times, but it was not effective” by “four times. Still, it was not effective”
Response: “the involvement of the formulation enhanced the efficacy by about four times. Still, it was not effective in killing the more tolerant weedy rice”.
Comments 3:
Figures 1C, 3D, and 4C could be colored to enhance visibility.
Response 3:
We have changed them colour. please confirm in the revision due to space limitation
1C,3D transfer to 4C,4C transfer 5C
Comments 4:
I believe that Figure 3A, which displays photosynthetic fluorescence parameters and leaf area, is not visually clear and could be improved. It is quite difficult to interpret due to its unclear appearance. Perhaps it would be better to present the data in a separate figure or improve its visual design. Alternatively, it could be more intense lines and symbols.
Response 4:
Thank you for your suggestion and we have presented Figure 3A with two separate figures, Figure 3 and Figure 4. to improve the unclear appearance. please confirm it in the revised version.
Comments 5:
Figures 6 B (1 and 4) could be colorful, while 2 and 3 could be more intense lines and symbols.
Response 5:
Thank you for your suggestion and we changed scatter to column diagram with different color, please check it in the revision. transfer to Figure 7 B ( 1 and 4 )

Reviewer 3 Report
Comments and Suggestions for Authors
The study by Zhang et al, investigated the effects of IMI herbicides on weedy rice, specifically focusing on the necrotic spotting phenotype. The necrotic spots primarily appeared in young seedlings within days after imazethapyr application. These spots later progress to necrosis. Severe cases resulted in extensive necrosis spread. Distinct differences were observed between the necrotic spotting phenotype of HRT1 and the conventional phenotype of HRT2, with the former displaying spotting without significant chlorosis while the latter exhibited gradual wilting. Population analysis revealed a shift in phenotype distribution over time, with the necrotic spotting phenotype becoming predominant initially but later transitioning to a mixed phenotype. Microscopic examination ruled out bacterial infection as the primary cause of necrotic spots, instead indicating vacuole-mediated necrosis as the underlying mechanism. Further experiments demonstrated that necrotic spotting was induced independently by imazethapyr and enhanced by the herbicide formulation, with higher concentrations causing necrosis even in resistant varieties. Photosynthetic fluorescence analysis revealed decreased efficiency and growth inhibition in response to herbicide concentration, with necrotic spots correlating negatively with photosynthetic efficiency. Transcriptome analysis identified differential expression pathways associated with carbon and nitrogen metabolism, photosynthesis, and ER stress, suggesting membrane structure damage and programmed necrosis as potential mechanisms. High temperature and strong light were found to exacerbate necrotic spotting. However, electron microscopy results contradicted the speculation that chloroplasts were the critical sites for necrosis development. Overall, the study provides insights into the mechanisms underlying herbicide-induced necrosis in weedy rice, highlighting the role of ER stress and cellular membrane damage in this phenomenon.
I have minor suggestions that can be easily addressed:
1. In Figure 1 and elsewhere, define CK.
2. Figure 1C, the x-axis can be relabelled treatment days.
3. Subscripts are not required when mentioning figure sections. Line 145, line 150, 163 and others can be corrected.
4. In figure 5C, rich ratio can be relabelled as enrichment ratio.
5. In discussion, from line 333, the reference's name should be mentioned just once.
6. In the supplementary text, spelling, numbering, and labeling of figures should be checked and figure legends can be elaborated for clarity.
Comments on the Quality of English Language
Minor editing required.
Author Response
Dear reviewer
Thank you very much for taking the time to review this manuscript. Please find the detailed responses below and the corresponding revisions/corrections highlighted/in track changes in the re-submitted files.
Comment 1:
- In Figure 1 and elsewhere, define CK.
Response 1:
We added the definition of CK at the charted
.” (A) Necrotic process, CK: water treated for control, EP: early phase, AP: later phase.”
Comment 2:
- Figure 1C, the x-axis can be relabelled treatment days.
Response 2:
We agree that d is unnecessary, so we removed it and color the figure.
Comment 3:
- Subscripts are not required when mentioning figure sections. Line 145, line 150, 163 and others can be corrected.
Response 3:
We removed the unnecessary subscripts and others we back to Text format, please confirm revision in line 142, 147, 161 et al.
Comment 4:
In figure 5C, rich ratio can be relabelled as enrichment ratio.
Response 4:
We have changed the ‘rich ratio’ to ‘enrichment ratio’ in Figure 6 in the revised version.
Comment 5:
In discussion, from line 333, the reference's name should be mentioned just once.
Response 5:
“Qian’s results were consistent with the above”, please confirm line 336
Comment 6:
- In the supplementary text, spelling, numbering, and labeling of figures should be checked and figure legends can be elaborated for clarity.
Response 6:
Thank you for your suggestion and we have checked the text, numbering, spelling and labelling of the supplementary Tables and Figures and corrected them in the revised version.
Sorry, this website can only upload one file at one time, so we changed the appendix, please.
